# Do we agree on who is playing the ball? Developing a video-based measurement for Shared Mental Models in tennis doubles

**Charlotte Raue** *, **Dennis Dreiskämper, Bernd Strauss**

Institute of Sport and Exercise Sciences, University of Muenster, Muenster, Germany

* charlotte.raue@wwu.de

**Data Availability Statement:** Data underlying the study are available on the OSF repository: https://osf.io/89f2e/.

## Abstract

Sport teams work in complex environments in which each member's tasks are mutually dependent on those of the others. To function effectively, expert teams generate Shared Mental Models (SMMs) to help adapt their own behavior to that of the others and master upcoming actions. Although SMMs have been discussed in domains such as organizations, there is still little research in the context of sport. One reason for this is that measurement methods have failed to incorporate the dynamic nature of the sport context. This study reports on the development of a video-based measurement of SMMs in tennis doubles. It examined the quality criteria first in a pilot and then in a main study. The final video-based measurement consists of 35 tennis doubles video clips requiring decisions on ball-taking behavior in two conditions (*Self* and *Partner*). In the condition *Self*, participants reported their own responses; in the condition *Partner*, those of their partner. The main study analyzed 29 male tennis teams with a mean age of 34.57 years ($SD$ = 12.25) and a mean of 22.79 years ($SD$ = 10.49) tennis experience. SMMs were analyzed for each partner as the inter-player agreement (*Self–Partner*) and averaged for each team. After completing the video-based measurement, participants filled out questionnaires on SMMs, team trust, and demographics. Results indicated that not only the split-half reliability ($r$ = .49), the content validity ($\eta_p^2$ = .23), the inter-player agreement ($r$ = .63), and the inter-player agreement and accuracy ($r$ = .61), but also the feasibility of the measurement were good. However, no relationships to the proposed convergent or criterial validity measures were found. In sum, measuring SMMs with a video-based test is possible and a promising method. No relationship to the frequently used questionnaires was found, suggesting that the two target different parts of SMMs. Future research should carefully examine and choose the appropriate measurement.

## Introduction

In tennis doubles teams, the coordination between two players is critical for deciding who is going to play the ball. If both rely on the other to play the ball, nobody plays it and the point is lost. If they coordinate their actions and agree as a team on deciding about who is playing the

**Funding:** The project was funded by the German Research Foundation (DFG) within the Research Training Group 1712-2 "Trust and communication in a digitized world.

**Competing interests:** The authors have declared that no competing interests exist.

ball, they can perform well [1, 2]. Traditionally, these decision-making processes have been researched on the individual level with different approaches, such as intuitive and deliberate decision-making [e.g., 3] or heuristic and bounded-rational decision-making [e.g., 4]. However, since these decisions for example in tennis double teams are formed within the team, scholars have long been interest in the notion of team decision making in terms of Shared Mental Models (SMMs) (e.g., [5–9]). SMMs are generally defined as "an organized understanding or mental representation of knowledge that is shared by team members" [10, p.123]. Based on these knowledge structures, their SMMs, team members form accurate explanations and expectations regarding the situation at hand and incorporate the demands of the task and the team members into their subsequent actions [11]. Thus, work on SMMs has incorporated aspects from the individual decision-making process, such as chunking the knowledge and facilitating a heuristic decision-route as a guideline for subsequent actions and the coordination of team members. But most importantly work has highlighted, that these knowledges structure should be shared across team members, to facilitate similar behaviour from the team and in turn implement effective coordination. Research in the organizational context has proven, that SMMs lead to better team performance [e.g., 12], however research adapting and replicating these results in the sport context is growing, but scarce. One reason for that being, that no universal measurement method exists [13] and the measurement methods lack the incorporation of the dynamic-sport environment [14]. Thus, the aim of the study is to develop and validate a sport-specific measurement method for SMMs.

## Theoretical background

Individual decision-making has often been classified on the continuum between intuitive and deliberate decision-making. While intuitive decision-making is classified as quick through perceiving patterns and acting upon their linked specific set of actions. Deliberate decision-making is often classified as the slower form, carefully considering all relevant information [3]. Both happens on the individual level, such that in the sport context athletes might make their decision intuitively or deliberately. However, if athletes are within a team, there are more things to contemplate than solving the task at hand, such as considering how their team members behave in order to coordinate their own actions with them. Thus, team decision-making has been often thought of in terms of SMMs, which is a team-level psychological state [8]. SMMs can be compared to intuitive decisions through the structured knowledge and patterns and furthermore, these are shared across team members. In general, SMMs can be differentiated on two different levels: (a) the mental models on the individual level and (b) the sharedness or similarity of those mental models on a group level [15].

On the individual level, mental models represent objects, actions, situations, or people [16]. They work as a chunk of all this knowledge and are built up through experience and observation [16]. The knowledge compressed into a mental model enables individuals to describe, predict, and explain the behavior at hand [17], similar to the intuitive decision-making process. The more experienced someone is within a certain context, the more details are worked into the mental models and thus, the more detailed the mental models are when compared to those of novices [11].

On the group level, SMMs assume that team members have similar mental models or share their mental models about upcoming actions [8]. This empowers team members to anticipate the needs and actions of others, and, through this mutual interpretation of the situation, "to be on the same page" regarding what is going to happen next [13]. This does not require team members to all have the same knowledge. Instead, some aspects related to their individual tasks need to be complementary (e.g., knowledge about blocking or defending in beach

volleyball; see [15]). However, in general team members need to share knowledge on the accurate decision for the right reason [18]. If all team members share inaccurate knowledge, they would exhibit the wrong behaviour and learning the correct knowledge is impeded [19]. Traditionally, SMMs are grouped into two broad categories: (a) task-related knowledge and (b) team-related knowledge [5]. Task-related knowledge includes performance requirements and goal strategies; team-related knowledge includes the personal team interaction requirements and the individual skills of team members [13]. An example of task-related knowledge in tennis doubles is anticipating where the ball is going to land in order to determine how to return. An example of team-related knowledge is letting the partner play the ball because she or he plays the stronger return.

Initially, SMMs were observed in nonsport environments such as organizational teams in which experts have to coordinate their behavior without the need for overt communication but rather by relying on a compatibility in members' cognitive understanding of key elements of their performance environment [11, 12]. Based on these observations, the focus was mainly on the SMMs–performance relationship and less on the SMMs–behavioral processes relationship. In two meta-analyses, DeChurch and Mesmer [12, 20] showed that SMMs have a strong positive relationship with both team behavioral processes (e.g., planning behaviors) and team performance (e.g., degree of task completion). They further noted that the way SMMs are measured moderates the SMMs–team behavioral process relationship [20]. For example, elicitation techniques all yielded positive relationships. Elicitation techniques captured the content of the mental models. Examples are similarity ratings (e.g., [21]) and questionnaires (e.g., Team Assessment Diagnostic Measure, TADM; [22]). These findings were also replicated in a recent study showing that SMMs predicted the adaptive team performance of undergraduate students in a simulated organizational context [23]. In this study, performance was measured with a card-sorting task and adaptive team performance was operationalized as the difference in performance between timepoints. SMMs were measured with averaged correlations across members in which all task attributes were provided and participants rated how related they were to the other attributes. Furthermore, recent studies have also confirmed that SMMs relate to other behavioral processes that are relevant for team performance (e.g., team trust). For instance, Guenther and colleagues [24] examined SMMs and team trust as part of a broader model using a 3-item scale to measure intragroup conflict and a 5-item scale for SMMs. They found that both SMMs and team trust were relevant for team coordination. In general, there is clear agreement on the importance of SMMs in effective organizational teams, but measuring SMMs remains complex and context-dependent, and there is no single, universal measurement method [13]. Although it is tempting to apply the results from work teams to the sport context, researchers need to take care when including the new context [25]. The sport context poses different problems compared to the organizational context. For example, in sport there are dynamic, rapidly changing, uncertain situations without much time to plan (e.g., the different rallies in tennis). If the results from the organizational context, however, are replicable within the sport context, SMMs are able to shed light on how expert teams' function and make decision within team sports.

## Measuring SMMs in sport

Still up to now, measurements in sport have been based on organizational measures such as specially developed items [6] and validated questionnaires [26, 27]. Moreover, interviews and document analysis have been conducted in order to include the specifics of the sport context [5, 28, 29]. These measures have delivered initial replications of the importance of SMMs in sport. For example, in line with the SMMs–team behavioral processes relationship, SMMs

relate to role clarity in elite ice hockey and handball players [6]. These authors operationalized SMMs with one general, one training-specific, and one opponent-specific SMM using specially developed items. In line with the SMMs–performance relationship, Filho and colleagues [27] found a positive relationship of SMM to collective efficacy and perceived performance in collegiate soccer players. These authors used the previously mentioned TADM questionnaire from the organizational context. However, the SMMs–performance relationship could not be replicated in an explorative multilevel analysis using the same measurement [26]. The authors argued that their sample was too homogeneous and therefore, the small variance was reflected in the nonreplicated result. Furthermore, the TADM questionnaire is not a sport-specific measurement. Sport-specific aspects have been included by conducting interviews with soccer players and soccer coaches and carrying out document analyses [5, 28, 29]. This resulted in a more sport-specific questionnaire called the Shared Mental Model in Team Sports Questionnaire (SMMTSQ; [30]). The SMMTSQ consists of three scales: general cognition, situational cognition, and efficacy beliefs. Those scales are further divided into 13 subscales and measured with 50 items (for further information, see [30]). Hence, to obtain a good SMMs measurement method in sport, it is necessary to address and incorporate the specifics of the sport context.

SMMs are knowledge chunks that act as guidelines for situational decisions. Therefore, when using questionnaires as a one-time measurement method for SMMs in the sport context, two basic problems emerge: First, questionnaires measure only an overarching, broad concept, which is present in that specific time point [14]. However, it is hard for team members to have that broad concept present within each situation they are facing. Therefore, the one-time questionnaire might not help to explain the coordination of team members in these various situations. Hence, a measurement method has to incorporate situational tendencies [14]. However, these problems could be addressed through applying a questionnaire on various occasions immediately after and before such situations have happened. Similarly, research for situational tendencies have been conducted with short interviews, immediately after points within a game [e.g., 31]. Second, using questionnaires or interviews within these situations still implies that team members know about their SMMs and can retrieve the knowledge they need to answer the questions deliberately and consciously. However, in the fast, dynamic sport environment with limited time for explicit planning [32], team members might have no knowledge of how their decision is guided. They normally have prior knowledge states regarding how actions are going to unfold [1, 8]. However, within the rapidly changing context, they need to update and adapt their SMMs continuously to fit the situations at hand and decide intuitively [33, 34]. This does not mean that their knowledge and expertise do not function as a guide within these situations. It rather suggests that athletes might not be able to recall how their action was influenced and are, therefore, unaware and unable to answer the questions appropriately. Within this line of reasoning, researchers would need to rethink how they measure these knowledge states and be careful on which part of the SMMs they are actually measuring. The differential access hypothesis about SMMs points out, that the SMMs are so complex, that different measurement method might measure different parts of SMMs [35]. Thus, applying questionnaire or interviews might measure the more deliberate part of SMMs. While, if the goal is to measure how the more intuitive SMMs across team partners, different measurement methods are needed. In their review of the current measurement methods for SMMs in sport [14], the authors concluded with a call to develop new measurement methods that incorporate indirect measures and also reflect the dynamic nature of sport by extending the well-established temporal occlusion paradigm (e.g., [36, 37]). This methodological approach might be the key to measure the more intuitive decision-making part of SMMs and facilitate research of SMMs in sport.

Currently temporal occlusion paradigms in sport are used mostly for expertise studies in individual athlete's decision-making. Within this paradigm, athletes watch an action (e.g., a

tennis hit) on videos, pictures, or in live actions. At certain timepoints (e.g., before, at, or after ball–racquet contact), the participant's view is occluded (e.g., by stopping the video sequence and masking the image). The use of this paradigm has consistently shown expertise differences in athletes. For example, a meta-analysis has shown that expert athletes can anticipate subsequent actions more accurately and at earlier timepoints than novices, and that they are more accurate in their decision making [38]. Hence, temporal occlusion paradigms are an effective measurement to access more intuitive decision-making within the dynamic sport environment. When now extending the temporal occlusion paradigm from the individual to the team level based on the theoretical background of SMMs, it facilitates to measure SMMs in dynamic situations that incorporate situation-specific and team-specific aspects. Thus, the adaptation of the temporal occlusion paradigm to the team setting can enrich our understanding of intuitive SMMs in sport teams and enhance the ecological validity of a sport specific SMMs measurement.

### Aim of the study

The aim of the present study was to address the aforementioned problems with prior measurements and develop and validate a situation-specific measurement for SMMs in sports. In line with the theoretical background of SMMs, we developed a video-based measurement for SMMs using the temporal occlusion paradigm to incorporate situational tendencies within the sport context. This responded to the call to extend the temporal occlusion paradigm from the individual to the team level [14]. In a first step, a pilot study served to identify and test the stimulus material, the setup, and the content validity on an individual level. Then, a main study was run to test feasibility, reliability, content validity as well as convergent and criterial validity on the team level.

## Pilot study: Identification and testing of stimulus material and setup

The aim of the pilot study was to test whether this measurement method is context-specific and able to measure individual mental models within one tennis player. The temporal occlusion paradigm was utilized as a basis and developed further to incorporate team-specific aspects such as deciding what the partner would do. The setup was in line with the mental model definition by Wilson [16] of *mental representations about objects, actions, situations, or people*. We then operationalized the mental model as the similarity of answers across conditions.

### Participants

A total of 20 intermediate-level male tennis players took part in the pilot study. Their mean age was 29.05 years ($SD = 8.76$) and they had 19.9 years of tennis experience ($SD = 6.87$). All participants actively played an average of 2.28 days ($SD = 1.13$) and 4.47 hours per week ($SD = 2.46$).

Participants had to be older than 18 years and play on at least an intermediate club level in Germany. Both criteria were set in order to ensure a level of expertise necessary for SMMs. Furthermore, only male players were recruited in order to match the players in the videos. Participants were recruited through flyers, social media, and snowball sampling. Interested participants were either tested at their training site or they could come to the university lab. All participants provided written informed consent before participation and the institutional review board of psychology and sport from the University of Muenster approved the study.

## Video measurement

The measurement instrument consisted of 35 videos depicting scenes from one whole match between two intermediate-level male tennis teams. The scenes ranged in difficulty (number of possible actions) and shots (where and which shots were taken). Recordings were taken from the back of the court positioned centrally to give a first-person impression (cf. [39]). The structure of the clips was standardized. After a lead-in to orient the participants, the rally started with a serve (first-person perspective), followed by a return (opponents' perspective) and a hit (first-person perspective). The following return by the opponents was temporally occluded at 80ms prior to ball contact and the screen immediately changed to the main task [40]. This time-point was chosen in line with earlier research suggesting that postural and contextual information are perceived. Thus, 80ms are in between pure anticipatory behavior (>140ms, [41]) and only reacting to the situation (at timepoint; [36]) and hint to more intuitive decisions. As a first step in testing the stimulus material, two experts evaluated the videos and agreed upon their appropriateness for intermediate players. Both experts had at least 30 years of tennis experience and played in higher leagues (3rd and 4th league in Germany; see, e.g., [39, 42, 43]).

The video measurement was programmed into OpenSesame as a button press task (cf. [44]). Each response cleared the screen for the next task. No performance feedback was given. All 35 videos were used in each of the following four conditions (as depicted in Fig 1):

Condition 1. Being the net player and deciding for oneself (*Net player–Self)* with the following instruction: Please put yourself in the position of the net player and indicate your own action as soon as the video stops.

Condition 2. Being the net player and deciding for the partner at the back (*Net player–Partner)* with the following instruction: Please put yourself in the position of the net player and indicate how your partner at the back will act as soon as the video stops.

Condition 3. Being the back player and deciding for oneself (*Back player–Self)* with the following instruction: Please put yourself in the position of the back player and indicate your own action as soon as the video stops.

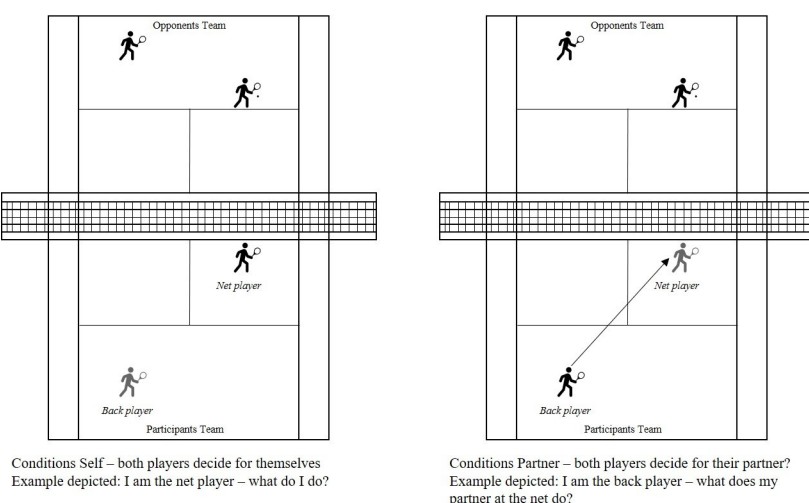

Conditions Self – both players decide for themselves
Example depicted: I am the net player – what do I do?

Conditions Partner – both players decide for their partner?
Example depicted: I am the back player – what does my partner at the net do?

**Fig 1. Illustration of task situation and the different conditions.**

Condition 4. Being the back player and deciding for the partner at the net (*Back player–Partner)* with the following instruction: Please put yourself in the position of the back player and indicate how your partner at the net will act as soon as the video stops.

Participants completed all four conditions. Before each of the four conditions, four warm-up videos were shown to familiarize participants with the measurement and enable them to adapt to each different condition (e.g., [43]). Although the four videos within the warm up remained the same, their order of presentation was randomized along with the sequence of conditions, and the 35 videos in each condition. Within each condition, the participants answered three questions on each video at the occlusion point; the first two in order to make inferences about their mental models and the last as a manipulation check to gain an indication regarding how much they knew about the situation. After the occlusion point, participants were first asked: "Will you/your partner play the ball?" (*Ball*) They used the keyboard to report either yes or no. Immediately afterward, they were asked "In which direction are you/your partner moving?" (*Movement)* Here as well, they indicated the direction on the keyboard. To facilitate a quick reaction, arrows were glued on the numpad (e.g. 8 –arrow up means moving up; 5 –no arrow means staying still). The third question "How sure are you about your decision?" (*Decision certainty*) was answered on a scale from 0 to 9.

## Procedure

Participants provided informed consent and were seated in front of the computer to start the self-paced video measurement. They received general instructions as well as a specific instruction before each condition, starting with the four warm-up trials. The same procedure was followed for all conditions. After completing the video measurement, participants completed a demographic questionnaire and received monetary compensation. After the study, participants were asked about the feasibility of the study.

## Data analysis

Data were analyzed with SPSS IBM 25.0 and Excel 16.29. The present analysis focused only on ball-taking behavior (*Ball*). Ball taking behavior was coded as 1 for yes and 0 for no. Ball-taking behavior in the condition *Net player–Partner* and *Back player–Self* was recoded so that all conditions were in the direction of the net player.

A percentage score of how participants decided was calculated for each video and each condition. In general, a high percentage meant a high level of agreement across players that the ball would be played. In contrast, a low percentage meant a high level of agreement across players that the ball would not be played. For further analysis, the extent of agreement is of interest (and not whether the ball is played or not). Thus, percentages under 50% were recoded to so that all agreements would be in the same direction. In general, the video stimuli should yield the whole range from low to high agreement. In order to check, whether the present stimuli did, we first clustered the video percentages in easy, medium and hard. Videos were clustered as easy, when participants agreed in all four conditions more than 75%. Videos were clustered as hard, when participants agreed in all four conditions less than 75%. All other videos were clustered as medium. Second, to check whether the clusters were appropriate, we calculated a repeated measures ANOVA across the three conditions.

The *Mental Model* was operationalized as a congruency in answers across conditions. Thus, we calculated correlations between each condition using the mean percentages. Normality was

tested using the Kolomogorov-Smirnov test, with the significance level set at .20. Because the condition *Net player–Self* was not normally distributed, we calculated Spearman's rho.

The feasibility of the study was identified through asking unstandardized open questions for feedback of the participants after completion of the study and to check whether problems emerged. Answers were grouped together with unstandardized observations of the experimenter during the pilot study. Problems which emerged half of the time or more were reported.

## Results: Pilot study

Table 1 shows the percentage scores on all videos within each condition as well as the overall mean and the standard deviation for each video. Six videos were clustered as easy videos (overall percentage of agreement: $M = 90.4\%$, $SD = 9.76$, 11 videos as medium (overall percentage of

**Table 1. Percentage agreement for all conditions on all videos.**

| Difficulty level | Video No. | Net player–Self | Net player–Partner | Back player–Self | Back player–Partner | Overall Mean | Overall SD |
|---|---|---|---|---|---|---|---|
| Easy | 2 | 85% | 95% | 90% | 100% | 93% | 6.45 |
| | 9 | 85% | 85% | 85% | 90% | 86% | 2.50 |
| | 12 | 100% | 95% | 90% | 100% | 96% | 4.79 |
| | 13 | 95% | 95% | 95% | 85% | 93% | 5.00 |
| | 23 | 95% | 85% | 90% | 85% | 89% | 4.79 |
| | 27 | 80% | 95% | 90% | 80% | 86% | 7.50 |
| Medium | 4 | 90% | 80% | 85% | 70% | 81% | 8.54 |
| | 14 | 80% | 75% | 65% | 85% | 76% | 8.54 |
| | 15 | 75% | 75% | 85% | 85% | 80% | 5.77 |
| | 19 | 85% | 85% | 85% | 70% | 81% | 7.50 |
| | 20 | 75% | 85% | 90% | 80% | 83% | 6.45 |
| | 25 | 80% | 55% | 70% | 50% | 64% | 13.77 |
| | 28 | 60% | 75% | 85% | 60% | 70% | 12.25 |
| | 29 | 85% | 90% | 80% | 65% | 80% | 10.80 |
| | 31 | 70% | 70% | 90% | 70% | 75% | 10.00 |
| | 33 | 70% | 85% | 85% | 85% | 81% | 7.50 |
| | 34 | 60% | 80% | 85% | 55% | 70% | 14.72 |
| Hard | 1 | 58% | 60% | 50% | 60% | 57% | 4.75 |
| | 3 | 55% | 60% | 50% | 60% | 56% | 4.79 |
| | 5 | 55% | 65% | 55% | 55% | 58% | 5.00 |
| | 6 | 70% | 55% | 55% | 75% | 64% | 10.31 |
| | 7 | 70% | 60% | 50% | 60% | 60% | 8.16 |
| | 8 | 75% | 65% | 60% | 65% | 66% | 6.29 |
| | 10 | 70% | 70% | 55% | 60% | 64% | 7.50 |
| | 11 | 65% | 55% | 65% | 60% | 61% | 4.79 |
| | 16 | 65% | 50% | 65% | 60% | 60% | 7.07 |
| | 17 | 70% | 70% | 50% | 75% | 66% | 11.09 |
| | 18 | 50% | 50% | 65% | 55% | 55% | 7.07 |
| | 21 | 65% | 65% | 65% | 70% | 66% | 2.50 |
| | 22 | 65% | 75% | 50% | 65% | 64% | 10.31 |
| | 24 | 50% | 65% | 65% | 60% | 60% | 7.07 |
| | 26 | 65% | 55% | 63% | 75% | 65% | 822 |
| | 30 | 65% | 60% | 55% | 53% | 58% | 5.50 |
| | 32 | 65% | 50% | 60% | 55% | 58% | 6.45 |
| | 35 | 50% | 55% | 60% | 55% | 55% | 4.08 |

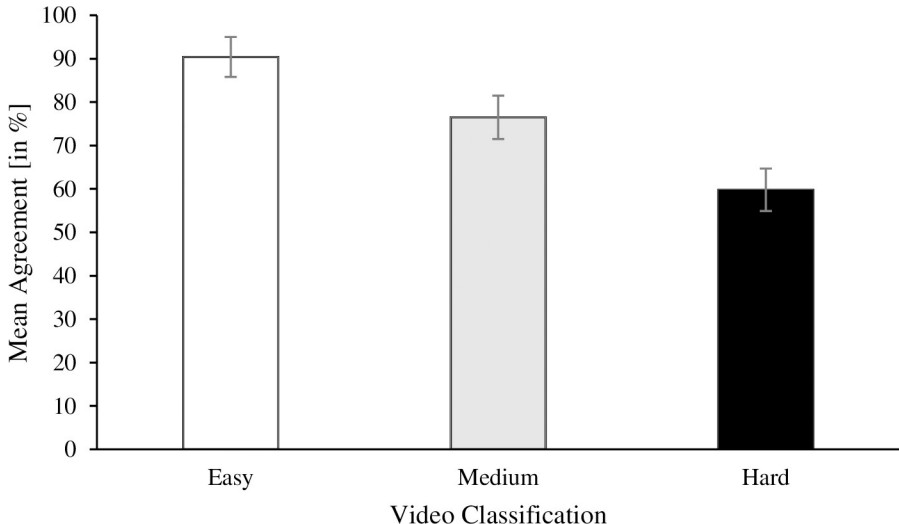

**Fig 2. Mean agreement on easy, medium and hard videos.** Note: Error bars resemble the 95% Confidence Intervals of the Mean.

agreement: $M = 76.5\%$, $SD = 10.45$), and 18 videos as hard (overall percentage of agreement: $M = 59.80\%$, $SD = 10.42$).

Fig 2 depicts the percentage of the mean agreement per video classification. Mauchly´s Test of Sphericity indicated that the assumption of sphericity had been violated, $\chi^2(2) = 10.172$, $p = .006$, and therefore a Greenhouse-Geisser correction was used. A repeated-measures ANOVA revealed a significant difference between easy, medium, and hard videos, $F(1.40, 26.54) = 68.54$, $p < .001$, $\eta_p^2 = .78$. Post hoc analyses with Bonferroni correction indicated that the percentages of agreement were higher on easy than on medium videos ($p < .001$, 95% CI [0.09, 0.16], $d = 1.86$) or hard videos ($p < .001$, 95% CI [0.24, 0.36], $d = 2.66$). Percentages for medium videos were higher than for hard videos ($p < .001$, 95% CI [0.08, 0.25], $d = 1.142$).

Table 2 reports the correlations for measuring *Mental Models* within each tennis player. All conditions correlated very highly ($.85 > r > .94$).

## Feasibility feedback

In general, participants found the study feasible, however two problems emerged: First, participants reported to have difficulty maintaining a high level of concentration throughout all four conditions and reported getting tired after a while. Second, participants noticed that the videos remained the same in every condition. Some were even able to recall their own decisions on previous videos and this influenced their decisions on the videos in the later conditions.

**Table 2. Correlational table across the four conditions.**

|  | Net player–Self | Net player–Partner | Back player–Self | Back player–Partner |
|---|---|---|---|---|
| Net player–Self | 1 | .895** | .853** | .899** |
| Net player–Partner |  | 1 | .935** | .888** |
| Back player–Self |  |  | 1 | .890** |
| Back player–Partner |  |  |  | 1 |

For the condition Net player–Self, we calculated Spearman's rho; for all others, Pearson's *r*.

** $p < .01$ (2-tailed).

### Interim discussion

The aim of the pilot study was to test whether the video measurement was appropriate and could measure individual mental model, before commencing to use the video measurement for shared mental models on the team level. A total of 35 video clips plus four warm-up trials were used as stimulus material. The results of the pilot study indicate that the videos vary in difficulty with significant differences between easy, medium, and hard videos. Hence, the stimulus material was appropriate [as recommended by 45]. Furthermore, the video test was able to measure the individual mental model, indicated through the high correlation across situations. For the main study, a team level shared mental model can be calculated for the net player (using conditions *Net player- Self/Back player–Partner*) and for the back player (*Back player–Self/Net player–Partner*). Lastly, the general video measurement seemed feasible for participants. However, based on the feedback about the difficulty in concentrating and, even more importantly, on recalling their own decision on prior videos, we decided to shorten the video measurement. First, we decided to only measure the SMM for the net player and thus, using only two conditions. Second, as the main study should measure more intuitive decisions rather than deliberate ones, we added a 3-s time limit for ball-taking behavior and movement directions to avoid participants being able to recall their actions and deliberately thinking about their decisions.(cf. [46]).

## Main study: Testing the quality criteria on the team level

The central aim of the main study was to measure intuitive SMMs within double tennis partners and replicate the video-based measurement gained on an individual level on the team level. The study followed a between-subject design. Based on the results of the pilot study, we analyzed SMMs for the net player. This was possible because in doubles tennis, both partners play at the net alternatively. Because a video-based measurement is still a newly developed method incorporating the fact that SMMs are inextricably tied to context [47], the overarching aim of the main study was to test the quality criteria of the measurement method. Therefore, we calculated split-half reliability, construct and content validity, correlational constructs, and convergent and criterial validity.

### Participants

Participation criteria were the same as in the pilot study with one addition: We recruited double tennis teams playing actively in the current season and stipulated that both team partners had to take part in the study at the same time. All participants provided written informed consent before participation and the institutional review board of psychology and sport from the University of Muenster approved the study. A total of 68 intermediate male tennis players (forming 34 double teams) agreed to take part in this study. Ten participants were excluded either because of technical difficulties while testing or because one of the team members could not adapt to the fast reaction time (max. 3 seconds) and thus, had only missing values. Because we aimed to measure SMMs on the team level, the whole team was excluded in these cases. On average, the final sample of 29 teams was aged 34.57 years ($SD$ = 12.25) and had 22.79 years of tennis experience ($SD$ = 10.49) and 17.65 years of double tennis experience ($SD$ = 11.05). Furthermore, on average the team partners player for 4.92 years ($SD$ = 7.06) together.

### Instruments

**Video measurement.**  In general, this was the same as in the pilot study, but now consisted of only two conditions: *Net player–Self* and *Back player–Partner* (see Fig 1). Furthermore, we

added a 3-s time limit to the variables *Ball* and *Movement*. *Decision certainty* had no time limit.

**Demographic questionnaire.** We collected several demographic data (e.g., age, individual tennis experience) and specific variables relevant for building SMMs (i.e., task experience, expertise, and team familiarity; e.g., [11]). Empirical research has shown that experience and team familiarity are linked to SMMs (e.g., [42, 48]). *Task experience* was measured with the item: "How long have you been playing doubles tennis (in years)?" *Expertise* was measured with: "What was your highest playing league?" *Team familiarity* was measured with: "How long have you been playing with your current partner (in years)?"

**Shared Mental Models in Team Sport Questionnaire (SMMTSQ).** As described above, the SMMTSQ consists of three scales with 13 subscales. The three scales are general cognition, situational cognition and collective efficacy. Because the video measurement is based on different situations, we used only the four situational subscales: Anticipation (4 items), Creativity (4 items), Experience (2 items), and Knowing each other's abilities (4 items). Although the experience subscale has only two items, we still included it in the present study. A two-item factor, is reliable if the items correlate highly and are relatively uncorrelated to the others [49], which was the case in the original manuscript [30]. The internal consistency of the originally reported scales was good with Cronbach's $\alpha = 0.84$, $\alpha = 0.86$, $\alpha = 0.79$, and $\alpha = 0.77$ respectively (see [30]).

**Trustworthiness instrument in sport.** Trust is a psychological state arising through the perceived trustworthiness of another individual [50, 51]. Perceived trustworthiness is composed of ability, benevolence, and integrity (for further elaboration, see [51]). Team trust is a key component for a team's coordination [52] and has been shown to be an antecedent of SMMs [2, 53]. Trust is assessed by measuring the perceived trustworthiness of the partner in terms of ability, benevolence, and integrity. The original trustworthiness measurement was developed for trust in management [54] and has been validated successfully in sport [55]. We used an adapted short version with three items for ability (e.g., My tennis partner is very competent in executing her or his tasks), three for benevolence (e.g., My tennis partner is very concerned about my welfare), and three for integrity (e.g., My tennis partner has a strong sense of justice). The internal consistency of the scales was appropriate with a Cronbach's alpha for perceived trustworthiness of $\alpha = 0.88$, perceived ability of $\alpha = 0.75$, perceived benevolence of $\alpha = 0.81$, and perceived integrity of $\alpha = 0.63$ (cf. [56]).

## Procedure

The general procedure was the same as in the pilot study, except that both players were measured simultaneously. Both players arrived together. After being welcomed, they were accompanied by two experimenters to two different locations. These locations were either two different rooms or within one room at a fair distance between partners. Participants were allowed to ask the experimenter questions during the warm-up trials, but were not allowed to speak with their partner. After providing informed consent and completing the video measurement, each participant filled out a demographic questionnaire, the short trustworthiness questionnaire, and the selected SMMSTQ subscales and received monetary compensation.

## Data analysis

For the main study, the analysis focused on the Shared Mental Model of *Ball*. Split-half reliability was calculated per person on all 35 videos to determine how reliably a person could attain the same or similar score when using the video measurement again. For this analysis, we included only the *Net player–Self* condition, because the *Back player–Partner* condition was

based on the partner and these scores can vary. For each participant, we calculated a mean across all 35 videos (overall). Furthermore, because all videos were shown randomly, we split them into the first 18 videos (Half 1) and the second 17 videos (Half 2).

For content validity our analysis was threefold, once on construct level regarding the inter-player agreement and the accuracy of this agreement, once on video-measurement level replicating the results of the pilot study. First, on construct level we assumed that the inter-player agreement about the net player behavior of both partners should be shaped similarly, in order to depict an overall Team SMM. Thus, we analyzed the correlation of the inter-player agreement of both players (SMM-A and SMM-B). Hereby, inter-player agreement was operationalized for each partner individually as an overall congruency score. It consists for Partner A (SMM-A) of the comparison of Ball $Self_{\text{Player A}}$ with Ball $Partner_{\text{Player B}}$; and for Partner B (SMM-B) of Ball $Self_{\text{Player B}}$ with Ball $Partner_{\text{Player A}}$. If, for example, in condition $Self_{\text{Player A}}$, the player said "I am going to play the ball" and the partner said in the condition $Partner_{\text{Player B}}$ "He is going to play the ball," answers were congruent. The answers were also congruent when both denied playing the ball. Both were recoded as a congruency score of 1. However, if their answers were not congruent, they were given a score of 0 for that video. This operationalization procedure was in line with prior research [23].

Second, on the construct level we examined whether team partners would agree on the accurate decision for the right reason [18]. As indicated in the pilot study, the video stimuli are quite ambiguous. We operationalized the accurate decision as the majority decision on the easy videos from the pilot study. Thus, if players would indicate to play the ball on videos 2,9,12 and indicate to not play the ball on videos 13,23,27 they would make an accurate decision. For each accurate decision we allocated one point and summed these points across partners. We then calculated the agreement between players as described above on these six videos. We assumed, that the accurate decision and the SMMs would correlate.

Third, on video-measurement level we operationalized *Team SMM* as an average of SMM-A and SMM-B. We replicated the results of the pilot study on the team level and determining whether *Team SMM* depended on the three difficulty levels of the videos (easy, medium, and hard) as well as the *Decision certainty* of the players.

As correlational constructs, we examined task experience, expertise, and team familiarity. Empirical research has shown that they are relevant for building SMMs in the first place (e.g., [11, 42]). Thus, we hypothesized that all three would predict *Team SMM*.

We tested convergent validity with the situational subscales from the SMMTSQ [30]. At this point, it should be stressed that these subscales might not be the best option for measuring convergent validity, because questionnaires are self-report measures and neither context-dependent nor situation-specific. Nonetheless, they are at least phrased in a task-specific way. As discussed previously, questionnaires are frequently used to measure SMMs and this questionnaire had at least been developed to be team-sport-specific. Hence, we calculated convergent validity, but expected only minor relationships to the video measurement.

Team trust was tested as criterial validity. Because trust was also measured with a questionnaire, similar measurement problems might arise. Hence, here as well, we expected only minor relationships to the video measurement.

In general, for all tests assuming normality, we calculated the Kolomogorov-Smirnov test, with the significance level set at .20. If lower, the assumption of normality was violated and we adjusted the measures appropriately. If not otherwise reported, normality can be assumed.

## Results: Main study

Table 3 reports the means and standard deviations for all variables.

**Table 3. Descriptives in the main study.**

| Quality criteria | Variables | N | M | SD | Min | Max |
|---|---|---|---|---|---|---|
| | Overall score | 57 | 0.53 | 0.15 | | |
| Split-half reliability | Half 1 | 57 | 0.55 | 0.17 | | |
| | Half 2 | 57 | 0.50 | 0.18 | | |
| | Team SMM _easy | 29 | 0.70 | 0.17 | 0.33 | 1.00 |
| | Team SMM_medium | 29 | 0.60 | 0.13 | 0.29 | 0.88 |
| Content validity | Team SMM_hard | 29 | 0.57 | 0.12 | 0.38 | 0.81 |
| | Decision certainty_easy | 29 | 7.34 | 0.60 | 5.79 | 8.29 |
| | Decision certainty_medium | 29 | 7.10 | 0.57 | 5.95 | 8.23 |
| | Decision certainty_hard | 29 | 6.87 | 0.61 | 5.67 | 8.03 |
| Inter-player | SMM_A | 28 | 0.62 | 0.09 | 0.35 | 0.81 |
| Agreement | SMM_B | 29 | 0.58 | 0.11 | 0.38 | 0.77 |
| | Accuracy | 28 | 9.71 | 1.56 | 5 | 12 |
| | SMMs_accuracy | 28 | 0.70 | 0.17 | 0.33 | 1.00 |
| | Task experience | 29 | 17.65 | 11.05 | 7.00 | 50.00 |
| Correlational | Expertise | 28 | 2.89 | 1.42 | 1.00 | 5.00 |
| Constructs | Team familiarity | 29 | 4.92 | 7.06 | 0.00 | 35.00 |
| | Team SMM | 29 | 0.60 | 0.09 | 2.00 | 5.00 |
| Convergent validity | Anticipation | 29 | 3.87 | 0.46 | 3.13 | 4.80 |
| | Creativity | 29 | 3.70 | 0.40 | 2.88 | 4.60 |
| | Experience | 29 | 4.30 | 0.45 | 3.50 | 5.00 |
| | Knowing each other's ability | 29 | 4.15 | 0.50 | 2.75 | 5.00 |
| | Perceived trustworthiness | 29 | 4.13 | 0.45 | 3.06 | 4.94 |
| | Perceived ability | 29 | 4.14 | 0.47 | 3.17 | 4.83 |
| Criterial validity | Perceived benevolence | 29 | 3.95 | 0.67 | 2.67 | 5.00 |
| | Perceived integrity | 29 | 4.30 | 0.55 | 2.50 | 5.00 |

**Split-half reliability.** Both Half 1 and Half 2 correlated, $r(57) = .49$, $p < .001$. Furthermore, results showed that both Half 1, $r(57) = .86$, $p < .001$, and Half 2, $r(57) = .86$, $p < .001$, correlated with the overall score.

**Content validity–construct level.** Results indicated that the SMM for both partners correlated, $r(28) = .63$, p $< .001$.

The accuracy value was not normally distributed, while the SMMs of the accuracy was, thus, Spearman's rho was calculated. Results indicated that both correlated $r(28) = .61$, $p = .001$.

**Content validity–difficulty level of videos.** *Team SMM*. A repeated measures ANOVA revealed a significant difference between easy, medium, and hard *Team SMM*, $F(2, 56) = 8.44$, $p = .001$, $\eta_p^2 = .23$. Post hoc analyses using Bonferroni corrections indicated that Team SMM was higher (= more congruency) on easy than on medium videos ($p = .02$, 95% CI [0.01, 0.18]; $d = 0.61$) and on hard videos ($p = .003$, 95% CI [0.04, 0.23]; $d = 0.83$). However, revealed no difference between medium and hard videos (95% CI [-0.05, 0.11]).

*Decision certainty*. A repeated measures ANOVA revealed a significant main effect of easy, medium, and hard *Decision certainties*, $F(2, 56) = 36.39$, $p < .001$, $\eta_p^2 = .57$. Post hoc analyses using Bonferroni corrections indicated that Decision certainty was higher on easy than on medium videos ($p = .001$, 95% CI [0.09, 0.39]; $d = 0.81$) or hard videos ($p < .001$, 95% CI [0.32, 0.62]; $d = 1.46$). Decision certainty was also higher on medium than on hard videos ($p < .001$, 95% CI [0.11, 0.35]; $d = 0.89$).

*Correlational constructs.* A multiple regression was conducted to see whether *Task experience*, *Expertise*, and *Team familiarity* predicted *Team SMM*. Using the enter method, no significant influence was found, $F_{(3, 24)} = 1.4$, $p > .05$, $R^2 = .15$, $R^2_{adj} = .04$, $f^2 = .18$. The only significant predictor was the intercept of the model.

**Convergent validity.** Only the subscales experiences and knowing each other´s abilities are not normally distributed. *Team SMM* did not correlate with either the SMMTSQ subscale anticipation, $r_{(29)} = .16$, $p > .05$, the subscale creativity, $r_{(29)} = .17$, $p > .05$, the subscale experience $r_{(29)} = -.28$, $p > .05$, or the subscale knowing each other's abilities, $r_{(29)} = -.15$, $p > .05$.

**Criterial validity.** Perceived trustworthiness, ability, benevolence and integrity were not normally distributed. We found no relationship between *Team SMM* and perceived trustworthiness, $r_{(29)} = -.09$, $p > .05$. *Team SMM* did not relate to the individual components perceived ability, $r_{(29)} = -.30$, $p = .12$; perceived benevolence, $r_{(29)} = .13$, $p > .05$; or perceived integrity, $r_{(29)} = .11$; $p > .05$.

**Further exploration.** As for convergent and criterial validity, we measured perceived trustworthiness and SMM with the similar method of questionnaires. We further explored whether perceived trustworthiness and SMMs related on those measurement methods.

Neither overall perceived trustworthiness, benevolence, or integrity correlated with any of the situational SMMTSQ subscales. However, perceived ability did correlate significantly with all four: anticipation ($r = .57$, $p < .01$), creativity ($r = .50$, $p = .01$), knowing each other's abilities ($r = .54$, $p < .01$), and experience ($r = .60$, $p < .01$).

## Discussion

Shared Mental Models (SMMs) as a team decision making process contribute to a positive performance outcome and team processes in work teams. Initial empirical results replicate this finding in sport teams. However, research on SMMs in sport is limited due to measurement difficulties, failing to incorporate dynamic and intuitive aspects and the lack of a general measurement method. The present study aimed to address this research gap by developing a video-based measurement for SMMs and examining its validity.

The measurement instrument developed was based on the premise that SMMs include task-related knowledge and team-related knowledge. Furthermore, we assumed that SMMs guide behavioral actions and facilitate the coordination of the players intuitively and are thus dynamic and situation-specific. Therefore, we used video scenes as stimulus material, and asked participants to rate their own behavior and that of their tennis partner. After piloting this measurement method, we replicated it with tennis doubles teams to check the quality criteria. Overall, split-half reliability indexes, content validity on measurement-level and on the construct level showed that answers were situation-specific and able to measure intuitive SMMs indirectly. In contrast to explicitly asking for the possible content of more deliberate SMMs by means of questionnaires or interviews, this indicates that it might be possible to advance this field of study by using indirect measurement methods. To the best of our knowledge, the present study is the first to operationalize SMMs within a dynamic video-based measurement.

Within the broader field of team cognition research, other studies have also used novel, more dynamic measurements. The theoretical rationale stems from dynamic systems approaches, where team cognition emerge within the situation itself [e.g., 57]. For example, one study used the temporal occlusion paradigm to look at team coordination through joint decisions about an upcoming action made by participants who were all watching the same situation, but from different angles [58]. Participants were teammates watching the scenes from their actual position, teammates watching from another position, or non-teammates. They

found that the coordinative decision from teammates watching from their own position was best. However, teammates watching from a different position performed better than non-teammates. This suggests that being familiar with the team and one's own capabilities are both relevant. The authors concluded that team familiarity and team knowledge were relevant. Future research could integrate the team coordination aspect by judging measurements of task and team knowledge from different angles. This could examine the SMMs–coordination relationship empirically. Another recent study investigated team cognition and incorporated more dynamic situations by examining shared knowledge with on-court communication in a real-life task [48]. Participants had to conduct a real-life soccer pass and evaluate their own actions and their teammates' actions both before and after the task. The participants' communication was audiotaped and analyzed. The authors found a trend toward a correlation indicating that when shared knowledge of the soccer players increased, verbal communication decreased. They concluded that "situation-specific shared understandings emerge with effective, situation-specific collective training" (p. 5). Future research could integrate our SMMs measurement across various situations with real-life tasks to see whether they facilitate a decrease in real-life communication. Both examples show how more dynamic measurements are emerging in recent times. These methodological developments might contribute to further develop the theoretical rational of SMMs with the predetermined knowledge structures, incorporating more aspects from the dynamic systems approaches, which focus mostly on the situation and the context at hand.

In the current measurement the split-half reliability indexes, the content validity on measurement level and on construct level of the video measurement were good, but we found no relationship with proposed measures for convergent and criterial validity. Due to the lack of other validated options, our measurements for convergent validity were twofold: First, we chose single item measures for previously related variables of task experience, expertise, and team familiarity (e.g., [2, 40]). Second, we examined the relationship to SMMs measured through situational subscales of the questionnaire SMMTSQ [30]. For criterial validity, we examined a questionnaire assessing team trust [55, 56]. Contrary to our hypotheses predicting only minor correlations, we found no correlations. In order to see whether this finding could be explained through the difference in the measurement method, we conducted an exploratory secondary analysis. Here, we examined the relationship of the questionnaire-based measures of SMMs and team trust, and we did find some correlations. We shall use this to discuss the lack of a relationship between the video-based measurement SMMs and the other variables.

With the video-based measurement, we aimed to incorporate the dynamic aspects of, in this case, tennis teams. SMMs were then operationalized through an indirect approach to how they influence their situational decisions. In contrast, questionnaires generally measure a rather broader concept that might be task-specific (e.g., tennis double matches), but rarely situation-specific (e.g., using a questionnaire to gain knowledge about a particular situation within one tennis double match). Thus, the lack of relationship can be argued in line with the differential access hypothesis, that questionnaires and the video-based measurement access different parts of the SMMs. Thus, once objectives of the measurement are different: situation-based versus task-based. This could also explain why a relationship between the variables was found when both were measured with questionnaires. Another reason for the lack of a relationship could lie in how situational decisions are influenced. The video-based measurement of SMMs assumes an indirect approach to measuring SMMs intuitively. Through the time limit set for deciding, participants need to answer quickly and cannot deliberately decide [e.g., 3]). In contrast, however, the subscales of the SMMTSQ were derived from interviews and the theoretical background of SMMs. This questionnaire might thus reflect the deliberate part of SMMs about what athletes and coaches think and explicitly know to be part of a Shared Mental

Model. The lack of a relationship could thus indicate that there is a difference between what is explicitly known to influence certain situations and what guides decisions rather implicitly. If both are manifestations with which to measure the latent construct of SMMs, then future research should either integrate the two measurement methods or clearly distinguish between the two.

## Limitations and perspectives

Although the video-based measurement is a promising method, it needs to be further explored in different contexts and validated. Within this study, we used a rather homogeneous sample of only 29 teams of intermediate-level tennis players. To obtain more differentiated results, the sample should be expanded to include novices as well as high-elite teams. A diverse range of teams would open up the possibility for greater variance in the teams and determining in which expertise level SMMs are found. A further increase in the sample size would also permit different analyses (such as multilevel analysis). The intermediate-level sample is sufficient for this initial exploration, and the effect sizes within content validities measures are appropriate. Furthermore, due to the specific initial requirement (such as athletes need to be at least on an intermediate level), sample recruiting was limited.

Furthermore, the video-based measurement aims to capture SMMs in a more situational and realistic setting. Although the use of videos helps to establish a more realistic setting, the participants were still seated in a laboratory in front of a computer rather than out on the court. Future studies could further increase ecological validity by showing videos in a virtual-reality environment in which participants could indicate where they want to look through their head movement. Another improvement could be for participants to actually conduct the behavior on a court rather than indicating it through a keyboard.

Based on the current results, we cannot generalize to actual coordination or performance. Nonetheless, we can strengthen the belief that SMMs are relevant for coordinative behavior within tennis dyads. Future research should carefully consider which measurement method to use and intent to incorporate dynamic aspects. Hereby, gathering more empirical data on the SMMs–performance and SMMs–behavioral process relationships. The presented video-based measurement serves as a promising method for measuring SMMs.

## Acknowledgments

We thank Stefanie Watzlawczyk for her assistance in preparing the study and data collection, Jonathan Harrow for the help in English Editing.

## Author Contributions

**Conceptualization:** Charlotte Raue.

**Data curation:** Charlotte Raue.

**Formal analysis:** Charlotte Raue, Dennis Dreiskämper, Bernd Strauss.

**Funding acquisition:** Bernd Strauss.

**Methodology:** Charlotte Raue, Dennis Dreiskämper.

**Supervision:** Dennis Dreiskämper, Bernd Strauss.

**Writing – original draft:** Charlotte Raue.

**Writing – review & editing:** Dennis Dreiskämper, Bernd Strauss.

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
