## [Decision Letter · Decision Letter 0]

1 Jul 2020

PONE-D-20-06372

Do we agree on who is playing the ball? Developing a video-based measurement for Shared Mental Models in tennis doubles

PLOS ONE

Dear Dr. Raue,

Thank you for submitting your manuscript to PLOS ONE. After careful consideration, we feel that it has merit but does not fully meet PLOS ONE’s publication criteria as it currently stands. Therefore, we invite you to submit a revised version of the manuscript that addresses the points raised during the review process.

Please note that the reviewers have indicated major limitations associated with the paper and that, should you decide to attempt the major corrections, the revised manuscript will be sent back to the reviewers for further consideration. I would encourage you to provide a detailed point by point response to their concerns, indicating what changes you have made or providing an alternative explanation. 

We look forward to receiving your revised manuscript.

Kind regards,

Professor Dominic Micklewright, PhD CPsychol PFHEA FBASES FACSM

Academic Editor

PLOS ONE

Reviewers' comments:

Reviewer's Responses to Questions

**Comments to the Author**

1. Is the manuscript technically sound, and do the data support the conclusions?

Reviewer #1: Yes

Reviewer #2: Partly

2. Has the statistical analysis been performed appropriately and rigorously? 

Reviewer #1: No

Reviewer #2: No

3. Have the authors made all data underlying the findings in their manuscript fully available?

Reviewer #1: No

Reviewer #2: No

4. Is the manuscript presented in an intelligible fashion and written in standard English?

Reviewer #1: Yes

Reviewer #2: No

5. Review Comments to the Author

Reviewer #1: I am positive about this paper. The paper is well-written and advances a new method to measure knowledge agreement in dyadic teams. Nevertheless, modifications are needed before the paper reaches its full potential. The feedback listed below is meant in a positive and constructive spirit.

Abstract: I would soften the claim that the questionnaires used and the video-method developed measure different constructs. They might measure different parts of the same complex construct (see differential access hypothesis in Cooke et al., 2000). This is just a suggestion though.

Cooke, N. J., Salas, E., Cannon‐Bowers, J. A., & Stout, R. J. (2000). Measuring team knowledge. Human Factors, 42, 151–173.

The tennis players might have agreed, which indeed suggests that they shared knowledge. However, they might agree on the wrong decision. Teammates need to make “the right decision for the right reason” (knowledge of what to do and why to do it; see Filho & Tenenbaum, 2020). Accordingly, it is also important to analyse whether the players have agreed on the best decision. Please include this information or offer a rebuttal to this point in the revised manuscript.

Filho, E., & Tenenbaum, G. (2020). Team mental models: Taxonomy, theory, and applied implications. In G. Tenenbaum & R. C. Eklund (Eds.), Handbook of sport psychology (4th ed., pp. 611-631). Wiley Publication.

It is hard to make sense of Table 1. A bar graph with 95% CI would be more telling the reader. Table 1 can be kept in the paper.

Pg. 17: You need at least three items to properly represent a given latent construct. As such, it is unclear why the factor “Experience” with only two items was included in the study. Please clarify and cite relevant academic sources as appropriate.

Pg. 18: If this information is available, please add for how long the players have been playing together as part of the same team, rather than the years of doubles tennis experience in general.

Pg. 19: I disagree with the claim that questionnaires “are neither context dependent nor situation-specific”. Questionnaires are both context dependent and situation specific and that is why they cannot be generalised to different populations and why statistical metrics (alpha, model fit such as CFI and chi-square) should be reported every time a new sample is studied. Please rephrase or offer a rebuttal to this point in the revised manuscript.

Pg. 20: Your argument that trust was tested as a divergent construct is not totally in line with your claim that trust antecedes SMM. Trust can be used as evidence of both convergent and construct validity, which are two sides of the same coin, but you need to make this argument clear in your revised manuscript or provide a different and well supported rationale.

In the Results section, please (a) Add range to Table 3; (b) Cohen’s d to all mean comparisons (i.e., ANOVAs). Moreover, intra-class correlation coefficients need to be computed and reported for the SMM self/partner and all other measures. These values can be added to Table 3. The construct SMM is measured at the team-level and simple correlations do not account for team-level variability. In the same vein, the multiple regression analysis reported on page 22 is not correct as it does not account for team-level variability. Please run a multi-level analysis and discuss these points in the revised manuscript. You usually need 15 dyads (https://www.frontiersin.org/articles/10.3389/fpsyg.2019.01067/full) to run a multi-level analysis and you have that in your data set. This is a nice study but without a multi-level analysis your paper will fall short of its potential.

From my reading, the first paragraph on page 24 reads more like a literature review and belongs in the Introduction.

Please expand on your argument that the video-based measurement of SMM “blocks explicit knowledge from influencing the decision”. That’s an interesting insight that needs to be explained further and backed-up with relevant citations.

Minor

Pg. 8, l. 158: Change to “was to address”

Pg. 17, l. 346: Change “good” to “appropriate”

Reviewer #2: This is an interesting paper about shared mental models in sport context, specifically in tennis doubles.

In general Introduction don’t support SMMs under a strong theoretical approach.

When you refer to dynamic nature of the sport context, are you referring to a complex systems approach? Maybe some of the Introduction could be related to dinamical systems approach more precisely.

When author refer to “Implementing such a coordination effectively depends crucially on cognitive factors (e.g.,[3]). Are you referring to decision making processes?

What about decision-making? Is SMMs a way to decide better? As I know, decision-making is also based on procedural knowledge, that seems to be related to SMMs. Some of the concepts of SMMs seems to be very near to cognitive psychology, but authors don’t position their study with a strong theoretical framework. As a reader, I would expect a more precise approach to SMMs from a general theory or approach to a specific view like SMMs. I think that SMMs are very related to decision-making from a cognitive approach but Introduction remain still vague. Authors could enhance their theoretical introduction with a more precise theoretical position about SMMs.

Lines 55-60 It seems to refer to procedural knowledge. Is this correct?

The parallel conceptualization of SMMs form organizational field to sport context is not clear enough for a reader. Authors should explain what about SMMs could contribute to sport context.

After a reading of the introduction, I only want to make a reflection about SMMs contribution in sport research. What SMMs research could add to the extant literature? Why is needed to study from SMMs approach? What SMMs add that is not explained by cognitive psychology approach, decision making processes or procedural knowledge?

The same conclusions could be made if we study decision-making process and procedural knowledge in tennis players like in other studies you have cited in your manuscript?

Authors state from line 123 to 131 weakness of questionnaires to asses some cognitive issues due to the static approach of questionnaire, but they forgot that the same could be assessed by interviews during the game (see McPherson et al. studies).

Later, authors talk about temporal occlusion paradigm, that is proper from a motor control field. What this add to the theoretical approach of the study.

As a general impression, it’s very difficult to have a strong position of authors about SMMs. They talk about cognitive psychology approach concepts like procedural knowledge or decision making, also points some concepts of ecological dynamics, and finally talk about motor control concepts like temporal occlusion. For a reader, it’s very difficult to found a theoretical rational that guides the reader through the paper.

I think that Introduction section should be improved.

In the pilot study, authors explain that use temporal occlusion. As I know, temporal occlusion is often used to determine pre-cues that affect to decision-making. What about temporal occlusion is applied to this study? As I read, this study only use a frame-stop 80 ms previous to a tennis hit. Is this a temporal occlusion use? Or is only a previous stop of the video.

Reading the procedure and the questions about the situations, it seems that authors try to assess the way to anticipate a hit, but this anticipation not always appear in tennis players. Also, this lack of anticipation is event more present with low or middle expertise players. When a intermediate player decide if play the ball on the net or wait depends strongly of the direction of the ball and the previous hit of their mate. It seems unclear this procedure. Maybe a deeper explanation and justify the decisions of the researchers would be acknowledged.

Regarding the order of the information on the pilot study, why do you explain first the protocol and later the participants? Why not following the structure of participants, method, procedure, results order?

Why do you use percentages to data analysis in the pilot study? Why not an Intraclass correlation coefficient to assess agreement between players.

What is the reason to classify videos between easy, medium or hard? What criteria were applied? More information of this way to classify actions is needed. This appear as a first time on Results of the pilot study. More information about this classification is needed on method section.

Procedure and method is difficult to follow with this actual description. New information about method appear even on results section and this is not easy to follow.

I have some doubts about the use of one-way ANOVA to test differences between conditions (easy, medium and hard). Why authors make a one-way ANOVA? Why is not better a repeated-measures ANOVA if they are comparing situations within-subjects?

Regarding to feasibility, which method has been applied to analyze information about it? It seems that there is a open question but nothing is said about the way to analyze this qualitative information. It is open to misinterpretation biased by researchers.

Interim discussion seems to be made by personal opinions of the researchers. This don’t help to have confidence about this preliminary step.

After reading the pilot study I can’t assume that validity evidence is reached by researchers about the procedure to assess SMMs in tennis doubles.

The same criticisms are related to the main study. Then, despite the main study seems to add more information about content validity, certainty, convergent and divergent validity, I think that explanations about the previous study are needed first.

6. PLOS authors have the option to publish the peer review history of their article (what does this mean?). If published, this will include your full peer review and any attached files.

Reviewer #1: **Yes: **Edson Filho

Reviewer #2: No

---

## [Author Response · Author response to Decision Letter 0]

10 Sep 2020

Reviewer #1: I am positive about this paper. The paper is well-written and advances a new method to measure knowledge agreement in dyadic teams. Nevertheless, modifications are needed before the paper reaches its full potential. The feedback listed below is meant in a positive and constructive spirit.

Thank you very much for the constructive Feedback. We appreciate the positive spirit, comments and suggestions. They indeed improved the manuscript. 

Abstract: I would soften the claim that the questionnaires used and the video-method developed measure different constructs. They might measure different parts of the same complex construct (see differential access hypothesis in Cooke et al., 2000). This is just a suggestion though.

Cooke, N. J., Salas, E., Cannon‐Bowers, J. A., & Stout, R. J. (2000). Measuring team knowledge. Human Factors, 42, 151–173.

In line with your suggestion, we have softened the claim and now wrote in the abstract (line 37ff):

 “No relationship to the frequently used questionnaires was found, suggesting that the two target different parts of SMMs.” 

Furthermore, we included the thought and explained it more thoroughly in the paper, e.g., line 190ff:

 “Within this line of reasoning, researchers would need to rethink how they measure these knowledge states and be careful on which part of the SMMs they are actually measuring. The differential access hypothesis about SMMs points out, that the SMMs are so complex, that different measurement method might measure different parts of SMMs [35].” 

The tennis players might have agreed, which indeed suggests that they shared knowledge. However, they might agree on the wrong decision. Teammates need to make “the right decision for the right reason” (knowledge of what to do and why to do it; see Filho & Tenenbaum, 2020). Accordingly, it is also important to analyse whether the players have agreed on the best decision. Please include this information or offer a rebuttal to this point in the revised manuscript.

Filho, E., & Tenenbaum, G. (2020). Team mental models: Taxonomy, theory, and applied implications. In G. Tenenbaum & R. C. Eklund (Eds.), Handbook of sport psychology (4th ed., pp. 611-631). Wiley Publication.

Thanks for the suggestion. We analyzed this information and now included it in the Manuscript as part of the main study. See e.g., the description in the data analysis (line 1175ff):

“Second, on the construct level we examined whether team partners would agree on the accurate decision for the right reason [18]. As indicated in the pilot study, the video stimuli are quite ambiguous. We operationalized the accurate decision as the majority decision on the easy videos from the pilot study. Thus, if players would indicate to play the ball on videos 2,9,12 and indicate to not play the ball on videos 13,23,27 they would make an accurate decision. For each accurate decision we allocated one point and summed these points across partners. We then calculated the agreement between players as described above on these six videos. We assumed, that the accurate decision and the SMMs would correlate.”

It is hard to make sense of Table 1. A bar graph with 95% CI would be more telling the reader. Table 1 can be kept in the paper.

Thanks for the suggestion, we put forth a bar graph as you suggested. It is depicted in figure 2.

Fig 2. Mean Agreement on easy, medium and hard videos.

Note: Error bars resemble the 95% Confidence Intervals of the Mean.

Pg. 17: You need at least three items to properly represent a given latent construct. As such, it is unclear why the factor “Experience” with only two items was included in the study. Please clarify and cite relevant academic sources as appropriate.

Changed it in the manuscript to (line 1102ff): 

"Although the experience subscale has only two items, we still included it in the present study. A two-item factor, is reliable if the items correlate highly and are relatively uncorrelated to the others [49], which was the case in the original manuscript [30]"

Pg. 18: If this information is available, please add for how long the players have been playing together as part of the same team, rather than the years of doubles tennis experience in general.

 For the Pilot Study, this information is not available as participants were not measured within their team, but as individual players. For the main study it is included in the "team familiarity", however, we also included it in the participants description with the following sentence (line 1081) 

"Furthermore, on average the team partners player for 4.92 years (SD = 7.06) together."

Pg. 19: I disagree with the claim that questionnaires “are neither context dependent nor situation-specific”. Questionnaires are both context dependent and situation specific and that is why they cannot be generalised to different populations and why statistical metrics (alpha, model fit such as CFI and chi-square) should be reported every time a new sample is studied. Please rephrase or offer a rebuttal to this point in the revised manuscript.

We agree, that the sentences can be misinterpreted. Therefore, we rephrased the argument to the following (see line 171ff): 

“Therefore, when using questionnaires as a one-time measurement method for SMMs in the sport context, two basic problems emerge: First, questionnaires measure only an overarching, broad concept, which is present in that specific time point [14]. However, it is hard for team members to have that broad concept present within each situation they are facing. Therefore, the one-time questionnaire might not help to explain the coordination of team members in these various situations. Hence, a measurement method has to incorporate situational tendencies [14]. However, these problems could be addressed through applying a questionnaire on various occasions immediately after and before such situations have happened. Similarly, research for situational tendencies have been conducted with short interviews, immediately after points within a game [e.g., 31].”

Pg. 20: Your argument that trust was tested as a divergent construct is not totally in line with your claim that trust antecedes SMM. Trust can be used as evidence of both convergent and construct validity, which are two sides of the same coin, but you need to make this argument clear in your revised manuscript or provide a different and well supported rationale.

 Indeed, that was misleading in the original version. Thus, we used trust as criterial validity instead of a divergent construct. 

In the Results section, please (a) Add range to Table 3; (b) Cohen’s d to all mean comparisons (i.e., ANOVAs). Moreover, intra-class correlation coefficients need to be computed and reported for the SMM self/partner and all other measures. These values can be added to Table 3. The construct SMM is measured at the team-level and simple correlations do not account for team-level variability. In the same vein, the multiple regression analysis reported on page 22 is not correct as it does not account for team-level variability. Please run a multi-level analysis and discuss these points in the revised manuscript. You usually need 15 dyads (https://www.frontiersin.org/articles/10.3389/fpsyg.2019.01067/full) to run a multi-level analysis and you have that in your data set. This is a nice study but without a multi-level analysis your paper will fall short of its potential.

Thank you for the hint for the missing data. A) Range is included, b) Partial eta Square remains to rmANOVA; Cohens d has been added to all mean comparison (line916ff), (line 1237ff); c) Similar suggestion came from Reviewer 2. We based our classification on the descriptive statistics as indicated by the percentage value, as they make it more transparent to the reader. Calculating the ICC with our data has some problems, because the ICC is dependent on variance and some of our videos (classified as easy) have very low variance. For example, video 2 shows very little variances. The percentages show an agreement from 85 - 100%, thus a very high ICC among the raters for this video was expected. The ICC value obtained however, is at .19 - thus, a poor ICC. The same happened for another “easy” video. Furthermore, since our data depicts only 1 and 0, it leaves little room for variance and especially, if mostly everyone indicates the same value. Thus, we think the percentages are the best option.

Regarding the comment to run a multi-level analysis, we answer according the content validity testing and then the multiple regression and some final words:

For the content validity testing: Our main goal was to indicate, that there were consistent similarities in answers from partner A to partner B - thus, we used a simple correlational testing. Based on your comment we also created a MLM to predict SMM, based on partner while nested in teams. This model explained significantly less variance (and had higher values for AIC, BIC, LogLikelihood). Does this issue stem from our incorrect interpretation of your comment? Could you elaborate on what you want us to do, or do you think for the content validity the simple correlation testing suffices?

For the multiple regression: We examined a mixed-model MLM design accounting for team-level variability. Similar to our reported multiple regression, the only significant model increase was the intercept only model. Thus, the overall effect of the predictors in the mixed model was negligible (both in p-value and variance explained (less than 0.005)). Therefore, the only truly relevant predictor was the intercept (the mean of SMMs). This effect was also present in the multiple regression we calculated. Thus, we have adapted the result section to reflect this and added the following sentence: "The only significant predictor was the intercept of the model. "Do you want us to include the multi-level design in the result section as well?

Furthermore, we think that our present data differs from the data in the article you suggested. The article examines 3-level Model in longitudinal studies - both of which we would argue to no have in our data. In our opinion we have only two levels (Individuals nested in teams). The sample size suggestions in the article for 2-level Models is 50 and for level 1 - 100 – which is more than we have (29 level 2 and 58 level 1). Furthermore, in a first attempt, we tried calculating the MLM - but the residual variance across teams is so low, that we cannot predict the SMM on team level based on the level 1 predictors (experience and expertise) and level 2 predictor (team familiarity). So, while we agree that calculating a multi-level analysis would be the superior statistical method in general, we do not see any possibility with our data. If you still think, that it would be possible, we would be very grateful if you could provide more information on how you think it would work. The suggestions for a multilevel analysis is however, included in the paper (see line 1385f).

From my reading, the first paragraph on page 24 reads more like a literature review and belongs in the Introduction.

Thanks for the suggestion, the paragraph is now shortened, including only the relevant aspects of the introduction without literature review (line 1293ff): 

“Shared Mental Models (SMMs) as a team decision making process contribute to a positive performance outcome and team processes in work teams. Initial empirical results replicate this finding in sport teams. However, research on SMMs in sport is limited due to measurement difficulties, failing to incorporate dynamic and intuitive aspects and the lack of a general measurement method. The present study aimed to address this research gap by developing a video-based measurement for SMMs and examining its validity. 

Please expand on your argument that the video-based measurement of SMM “blocks explicit knowledge from influencing the decision”. That’s an interesting insight that needs to be explained further and backed-up with relevant citations.

Based on your comment, we rephrased the argument to the following (line 1378ff):

”The video-based measurement of SMMs assumes an indirect approach to measuring SMMs intuitively. Through the time limit set for deciding, participants need to answer quickly and cannot deliberately decide [e.g., 3]).” 

Minor

Pg. 8, l. 158: Change to “was to address”

Corrected. Thanks.

Pg. 17, l. 346: Change “good” to “appropriate”

Corrected. Thanks.

Reviewer #2: This is an interesting paper about shared mental models in sport context, specifically in tennis doubles.

Thank you very much for the kind words. We appreciate the comments and the suggestions and we think, that they improved the manuscript.

In general Introduction don’t support SMMs under a strong theoretical approach.

 Based on your feedback we rewrote the majority of the introduction, to reflect the decision-making approach more. See for example the first two pages of the revised manuscript.

When you refer to dynamic nature of the sport context, are you referring to a complex systems approach? Maybe some of the Introduction could be related to dinamical systems approach more precisely.

Thanks for the recommendation, we agree that there exists a link between SMMs and the dynamic system approach, which however, needs to be further researched in the future. Shared Mental Models and dynamic system approach have their roots on different sides.

Shared Mental Models are a form of team decision making, which has its root in individual knowledge structure dependent on prior knowledge and experiences. This is comparable to more intuitive decisions based on the structured knowledge (compare page 4 in the revised manuscript), where the decision roots more on the individual level.

On the other side, dynamic systems focus primarily on the context and situations and the decision root more on the interaction of the team members. Both views are important however, follow a different theoretical rationale. As we describe in the paper, we currently lack a fine measurement for the situational SMMs based on the individual. This study, aims to change that. 

However, we agree that contextual cues are very important and should be integrated in further measurements. While designing our study, we focused on SMMs as a theoretical background. However, we discussed more dynamic measurements in the discussion (compare line 1302ff). 

“Within the broader field of team cognition research, other studies have also used novel, more dynamic measurements. The theoretical rationale stems from dynamic systems approaches, where team cognition emerge within the situation itself [e.g., 57].”

And based on your comment, we adapted the discussion to highlight that further methodological developments are needed. (compare line 1334ff). And explicitly linked it to the dynamic system approach. 

“Both examples show how more dynamic measurements are emerging in recent times. These methodological developments might contribute to further develop the theoretical rational of SMMs with the predetermined knowledge structures, incorporating more aspects from the dynamic systems approaches, which focus mostly on the situation and the context at hand.”

When author refer to “Implementing such a coordination effectively depends crucially on cognitive factors (e.g.,[3]). Are you referring to decision making processes?

Indeed, we were, however, your feedback shows, that we should be more specific. We rephrased it to the following:

If they coordinate their actions and agree as a team on deciding about who is playing the ball, they can perform well [1,2]. Traditionally, these decision-making processes have been researched on the individual level with different approaches, such as intuitive and deliberate decision-making [e.g., 3] or heuristic and bounded-rational decision-making [e.g., 4]. However, since these decisions for example in tennis double teams are formed within the team, scholars have long been interest in the notion of team decision making in terms of Shared Mental Models (SMMs) (e.g., [5–9]).

What about decision-making? Is SMMs a way to decide better? As I know, decision-making is also based on procedural knowledge, that seems to be related to SMMs. Some of the concepts of SMMs seems to be very near to cognitive psychology, but authors don’t position their study with a strong theoretical framework. As a reader, I would expect a more precise approach to SMMs from a general theory or approach to a specific view like SMMs. I think that SMMs are very related to decision-making from a cognitive approach but Introduction remain still vague. Authors could enhance their theoretical introduction with a more precise theoretical position about SMMs.

Lines 55-60 It seems to refer to procedural knowledge. Is this correct?

 If SMMs are present in teams, their decisions are superior to teams without SMMs in the sense, that they can coordinate themselves better. So, SMMs are related to decision-making, because they function as team decision making processes. So procedural knowledge is part of SMMs in terms of task-related knowledge (see line 110ff). But in order to address your more general comments to approach SMMs from a general theory, we rewrote the introduction to depict the more general decision-making processes of intuitive and deliberate decision-making on the individual level. Based on these backgrounds we went on the team decision making processes present in SMMs. Please, see the first two pages.

The parallel conceptualization of SMMs form organizational field to sport context is not clear enough for a reader. Authors should explain what about SMMs could contribute to sport context.

 Thank you for the recommendation, we thus, added the following (line 140ff): 

“Although it is tempting to apply the results from work teams to the sport context, researchers need to take care when including the new context [25]. The sport context poses different problems compared to the organizational context. For example, in sport there are dynamic, rapidly changing, uncertain situations without much time to plan (e.g., the different rallies in tennis). If the results from the organizational context, however, are replicable within the sport context, SMMs are able to shed light on how expert teams’ function and make decision within team sports.”

After a reading of the introduction, I only want to make a reflection about SMMs contribution in sport research. What SMMs research could add to the extant literature? Why is needed to study from SMMs approach? What SMMs add that is not explained by cognitive psychology approach, decision making processes or procedural knowledge?

The same conclusions could be made if we study decision-making process and procedural knowledge in tennis players like in other studies you have cited in your manuscript?

 SMMs incorporates the decision-making processes on the individual, however, goes a step beyond in order to explain how those individuals are able to work well and coordinate themselves together in the team. In line with the saying “the sum is more than its parts”, a team as a whole has more to offer than adding only the individuals. While acknowledging the individual decision-making processes, SMMs specify that the team functions well together, if these knowledge structures (which are the basis for the decision) are shared across team members. While only examining the decision-making of individuals, we miss the opportunity to find out – what makes a team special. Thus, the team decision-making process is at stake or the SMMs. The other studies cited in the manuscript, do not measure SMMs as an intuitive decision-process and examined the agreement across players on those decisions. This is mostly due to the measurement difficulties currently present in SMMs research. Here our study aims to show a different way and a different measurement method for SMMs.

Authors state from line 123 to 131 weakness of questionnaires to asses some cognitive issues due to the static approach of questionnaire, but they forgot that the same could be assessed by interviews during the game (see McPherson et al. studies).

 Indeed, thank you for the suggestion, which we incorporated in the manuscript. Based on your comment and the comment from Reviewer 1 we rewrote most of the paragraph (line 172ff): 

“Therefore, when using questionnaires as a one-time measurement method for SMMs in the sport context, two basic problems emerge: First, questionnaires measure only an overarching, broad concept, which is present in that specific time point [14]. However, it is hard for team members to have that broad concept present within each situation they are facing. Therefore, the one-time questionnaire might not help to explain the coordination of team members in these various situations. Hence, a measurement method has to incorporate situational tendencies [14]. However, these problems could be addressed through applying a questionnaire on various occasions immediately after and before such situations have happened. Similarly, research for situational tendencies have been conducted with short interviews, immediately after points within a game [e.g., 31].”

Later, authors talk about temporal occlusion paradigm, that is proper from a motor control field. What this add to the theoretical approach of the study.

 Thanks for the question, to answer your comment, we added this to the manuscript (line 206ff):

“The differential access hypothesis about SMMs points out, that the SMMs are so complex, that different measurement method might measure different parts of SMMs [35]. Thus, applying questionnaire or interviews might measure the more deliberate part of SMMs. While, if the goal is to measure how the more intuitive SMMs across team partners, different measurement methods are needed. In their review of the current measurement methods for SMMs in sport [14], the authors concluded with a call to develop new measurement methods that incorporate indirect measures and also reflect the dynamic nature of sport by extending the well-established temporal occlusion paradigm (e.g.,[36,37]). This methodological approach might be the key to measure the more intuitive decision-making part of SMMs and facilitate research of SMMs in sport.”

As a general impression, it’s very difficult to have a strong position of authors about SMMs. They talk about cognitive psychology approach concepts like procedural knowledge or decision making, also points some concepts of ecological dynamics, and finally talk about motor control concepts like temporal occlusion. For a reader, it’s very difficult to found a theoretical rational that guides the reader through the paper.

I think that Introduction section should be improved.

We rewrote the Introduction to follow the theoretical rational throughout the paper and have a clearer rationale. We think that we have addressed all comments and we hope that we meet the ideas of the reviewer.

In the pilot study, authors explain that use temporal occlusion. As I know, temporal occlusion is often used to determine pre-cues that affect to decision-making. What about temporal occlusion is applied to this study? As I read, this study only use a frame-stop 80 ms previous to a tennis hit. Is this a temporal occlusion use? Or is only a previous stop of the video.

 Indeed, temporal occlusion is based on individual decision-making. In general, there is a difference between temporal and spatial occlusion paradigm in sport sciences. Both are used traditionally to see how they affect decision making. In spatial occlusion a certain area within the video is occluded (e.g., the racket in basketball players - see Hagemann, Canal-Brouwland, Strauss, 2006), in order to see whether this area is relevant for the decision of the athlete. Similarly, in temporal occlusion the video stops at a designated time-point (e.g., 80ms see our study) in order to see how that affects the decision. Depending on the research question there might be multiple temporal occlusion points necessary. However, in our study only one occlusion point was relevant, thus, you can call it a previous stop of the video or one temporal occlusion time-point.

Reading the procedure and the questions about the situations, it seems that authors try to assess the way to anticipate a hit, but this anticipation not always appear in tennis players. Also, this lack of anticipation is event more present with low or middle expertise players. When a intermediate player decide if play the ball on the net or wait depends strongly of the direction of the ball and the previous hit of their mate. It seems unclear this procedure. Maybe a deeper explanation and justify the decisions of the researchers would be acknowledged.

Thank you for the critical comment and providing an example for your rationale. We did want to measure more intuitive decisions and not only the anticipation. Thus, in order to justify our procedure, we added the following (line 278ff): 

“This time-point was chosen in line with earlier research suggesting that postural and contextual information are perceived. Thus, 80ms are in between pure anticipatory behavior (>140ms, [41]) and only reacting to the situation (at timepoint; [36]) and hint to more intuitive decisions.”

Furthermore, we set the selection criteria for the participants (to have an intermediate level - see line 263) in order to ensure a certain level of expertise and we had two experts acknowledging, that intermediate players are indeed able to decide upon those videos and the selected occlusion time-point, as they are very similar to a typical competition. However, we acknowledged your comment as well through expanding the limitation part reflecting that only using intermediate players limit the procedure (line 1384ff). 

"A diverse range of teams would open up the possibility for greater variance in the teams and determining in which expertise level SMMs are found."

Regarding the order of the information on the pilot study, why do you explain first the protocol and later the participants? Why not following the structure of participants, method, procedure, results order?

We rearranged the order on the basis of your suggestion. Some articles use a different order and prefer to first explain the measurements. However, it should not be confusing - thus, thank you.

Why do you use percentages to data analysis in the pilot study? Why not an Intraclass correlation coefficient to assess agreement between players.

We based our classification on the descriptive statistics as indicated by the percentage value, as they make it more transparent to the reader. Calculating the ICC with our data has some problems, because the ICC is dependent on variance and some of our videos (classified as easy) have very low variance. For example, video 2 shows very little variances. The percentages show an agreement from 85 - 100%, thus a very high ICC among the raters for this video was expected. The ICC value obtained however, is at .19 - thus, a poor ICC. The same happened for another “easy” video. Furthermore, since our data depicts only 1 and 0, it leaves little room for variance and especially, if mostly everyone indicates the same value. Thus, we think the percentages are the best option.

What is the reason to classify videos between easy, medium or hard? What criteria were applied? More information of this way to classify actions is needed. This appear as a first time on Results of the pilot study. More information about this classification is needed on method section.

We agree, that this information is missing. In general, the video stimuli should yield the whole range from low to high agreement, in order to count as a reliable measurement tool. We included that information in the interim discussion (line 1026f):

“Hence, the stimulus material was appropriate [as recommended by 45]”

Furthermore, we rewrote the manuscript in order to resemble the criteria in more detail (line 333ff): 

“A percentage score of how participants decided was calculated for each video and each condition. In general, a high percentage meant a high level of agreement across players that the ball would be played. In contrast, a low percentage meant a high level of agreement across players that the ball would not be played. For further analysis, the extent of agreement was of interest (and not whether the ball is played or not). Thus, percentages under 50% were recoded to so that all agreements would be in the same direction. In general, the video stimuli should yield the whole range from low to high agreement. In order to check, whether the present stimuli did, we first clustered the video percentages in easy, medium and hard. Videos were clustered as easy, when participants agreed in all four conditions more than 75%. Videos were clustered as hard, when participants agreed in all four conditions less than 75%. All other videos were clustered as medium. Second, to check whether the clusters were appropriate, we calculated a repeated measures ANOVA across the three conditions.”

Procedure and method is difficult to follow with this actual description. New information about method appear even on results section and this is not easy to follow.

 Thanks for the comment. We correct that, and rewrote these sections and included everything in the data-analysis section. However, if we tested assumptions necessary to conduct a certain statistical analysis, we left that in the result section. If you feel, that we missed something else, we would highly appreciate you pointing us to which information is new.

I have some doubts about the use of one-way ANOVA to test differences between conditions (easy, medium and hard). Why authors make a one-way ANOVA? Why is not better a repeated-measures ANOVA if they are comparing situations within-subjects?

 Thank you so much. That was a mistake, we now conducted a repeated- measures ANOVA as we did in the main study! We now corrected it for the pilot study, stating (line 908ff): 

“Mauchly´s Test of Sphericity indicated that the assumption of sphericity had been violated, χ2(2) = 10.172, p = .006, and therefore a Greenhouse-Geisser correction was used. A repeated-measures ANOVA revealed a significant difference between easy, medium, and hard videos, F(1.40, 26.54) = 68.54, p < .001, ηp2 = .78. Post hoc analyses with Bonferroni correction indicated that the percentages of agreement were higher on easy than on medium videos (p < .001, 95% CI [0.09, 0.16], d = 1.86) or hard videos (p < .001, 95% CI [0.24, 0.36], d = 2.66). Percentages for medium videos were higher than for hard videos (p < .001, 95% CI [0.08, 0.25], d = 1.142).”

Regarding to feasibility, which method has been applied to analyze information about it? It seems that there is a open question but nothing is said about the way to analyze this qualitative information. It is open to misinterpretation biased by researchers.

As this was a pilot-study we openly asked the participants afterwards, what they thought about the study and whether it was appropriate for them or if they had suggestions on what we could enhance for the main study. Unfortunately, we did not write that rigorously down and qualitatively analysed what the participants answered. We just noted down general issues, which we could address for the main study. Thus, indeed it is open to misinterpretation. However, as it was the pilot study, in order to test how suitable the measurement for tennis players is, we still find the grouped answers very helpful to improve the main study.

We addressed your comment in the revised version in the following way. First, we included the following line in the procedure section (line 325f) 

“After the study, participants were asked about the feasibility of the study.” 

Furthermore, we described the aforementioned procedure honestly in the data analysis section (line 366 ff): 

“The feasibility of the study was asked through unstandardized open questions, to gain information upon the experience of the participants and checking whether problems emerged. General issues were noted down.”

Interim discussion seems to be made by personal opinions of the researchers. This don’t help to have confidence about this preliminary step.

We rewrote the Interim Discussion in order to reflect and justify how we decided. Hopefully this helps building more confidence in the Pilot Study. Please, keep in mind, that it is only a pilot study and the main study should add the validity. The interim discussion now reads (line 1026ff) 

“The aim of the pilot study was to test whether the video measurement was appropriate and could measure individual mental model, before commencing to use the video measurement for shared mental models on the team level. A total of 35 video clips plus four warm-up trials were used as stimulus material. The results of the pilot study indicate that the videos vary in difficulty with significant differences between easy, medium, and hard videos. Hence, the stimulus material was appropriate [as recommended by 45]. Furthermore, the video test was able to measure the individual mental model, indicated through the high correlation across situations. For the main study, a team level shared mental model can be calculated for the net player (using conditions Net player- Self/Back player – Partner) and for the back player (Back player – Self/Net player – Partner). Lastly, the general video measurement seemed feasible for participants. However, based on the feedback about the difficulty in concentrating and, even more importantly, on recalling their own decision on prior videos, we decided to shorten the video measurement. First, we decided to only measure the SMM for the net player and thus, using only two conditions. Second, as the main study should measure more intuitive decisions rather than deliberate ones, we added a 3-s time limit for ball-taking behavior and movement directions to avoid participants being able to recall their actions and deliberately thinking about their decisions.(cf. [46]).“

After reading the pilot study I can’t assume that validity evidence is reached by researchers about the procedure to assess SMMs in tennis doubles.

The pilot study did not set out to measure shared mental models already. Based on the conceptualization of shared mental models (having knowledge structure present in each individual and then sharing these knowledge structures across team members), we aimed to examine with the pilot study whether we could indirectly measure these knowledge structures (through intuitively deciding on what to do next). Thus, the pilot study did not set out to meet all validity criteria. These were administered only to the main study. We therefore are still of the opinion that our aim for the pilot study is met. If you still disagree, could you help us in understanding your point and be more specific on what you are missing exactly?

The same criticisms are related to the main study. Then, despite the main study seems to add more information about content validity, certainty, convergent and divergent validity, I think that explanations about the previous study are needed first.

 We rewrote and restructured the pilot study based on your comments (and also as indicated in the answers above). We furthermore also adapted the main study based on the critics provided. We hope to convince the reviewer with our changes and additions.

---

## [Decision Letter · Decision Letter 1]

14 Oct 2020

PONE-D-20-06372R1

Do we agree on who is playing the ball? Developing a video-based measurement for Shared Mental Models in tennis doubles

PLOS ONE

Dear Dr. Raue,

Thank you for submitting your revised manuscript to PLOS ONE. After careful consideration, there is a relatively minor issue raised by the second reviewed that I would like to invite you to respond to.

We look forward to receiving your revised manuscript.

Kind regards,

Professor Dominic Micklewright, PhD CPsychol PFHEA FBASES FACSM

Academic Editor

PLOS ONE

Reviewers' comments:

Reviewer's Responses to Questions

**Comments to the Author**

1. If the authors have adequately addressed your comments raised in a previous round of review and you feel that this manuscript is now acceptable for publication, you may indicate that here to bypass the “Comments to the Author” section, enter your conflict of interest statement in the “Confidential to Editor” section, and submit your "Accept" recommendation.

Reviewer #1: All comments have been addressed

Reviewer #2: All comments have been addressed

2. Is the manuscript technically sound, and do the data support the conclusions?

Reviewer #1: Yes

Reviewer #2: Yes

3. Has the statistical analysis been performed appropriately and rigorously? 

Reviewer #1: Yes

Reviewer #2: Yes

4. Have the authors made all data underlying the findings in their manuscript fully available?

Reviewer #1: Yes

Reviewer #2: Yes

5. Is the manuscript presented in an intelligible fashion and written in standard English?

Reviewer #1: Yes

Reviewer #2: Yes

6. Review Comments to the Author

Reviewer #1: I commend the authors for a well done review and an interesting paper that has certainly contribute to the literature.

Reviewer #2: I acknowledge the response to my comments and the changes made by authors. The manuscript, in the current form, is near to be accepted.

Only remain an aspect that I consider that should be improved.

This aspect is related to the qualitative assessment of the pilot study. As I mentioned previously, I think that the assessment and data analysis is not accurate and still vague and not enough objective. Also, I was concerned about the effect of this qualitative assessment to the main experiment. The qualitative assessment highlight three main critiques. In this way, how authors modified the pilot study based on these critiques? Another question is related to the first critique made by the players, where players stated warm-up as a positive issue. Why is it considered as a critique?

Maybe if a quotation of a player is representative of each criticism it could be added to the manuscript.

overall, I have to congratulate authors by their good job.

7. PLOS authors have the option to publish the peer review history of their article (what does this mean?). If published, this will include your full peer review and any attached files.

Reviewer #1: **Yes: **Edson Filho

Reviewer #2: No

---

## [Author Response · Author response to Decision Letter 1]

5 Nov 2020

Reviewer #1: I commend the authors for a well done review and an interesting paper that has certainly contribute to the literature.

 Thank you so much! That is so nice to hear and makes us happy.

Reviewer #2: I acknowledge the response to my comments and the changes made by authors. The manuscript, in the current form, is near to be accepted.

Only remain an aspect that I consider that should be improved.

This aspect is related to the qualitative assessment of the pilot study. As I mentioned previously, I think that the assessment and data analysis is not accurate and still vague and not enough objective. Also, I was concerned about the effect of this qualitative assessment to the main experiment. The qualitative assessment highlight three main critiques. In this way, how authors modified the pilot study based on these critiques? Another question is related to the first critique made by the players, where players stated warm-up as a positive issue. Why is it considered as a critique?

Maybe if a quotation of a player is representative of each criticism it could be added to the manuscript.

overall, I have to congratulate authors by their good job.

 First, thank you so much, for the compliments. We are really happy to hear that, however, we see your concern and we are sorry, that our changes last time weren’t enough. 

Due to the length of the whole manuscript we previously shorten that point. However, to address your comment and analyse this section more objectively we looked at the descriptives of how many participants reported that. We included this in the data analysis section. We furthermore, removed the critique with the warm-up videos. As we now state to focus only on problems. Furthermore, we changed the title of “feasibility” to “feasibility feedback” in order to prevent misleading the reader.

(Data Analysis Section (line 312-316)

The feasibility of the study was identified through asking unstandardized open questions for feedback of the participants after completion of the study and to check whether problems emerged. Answers were grouped together with unstandardized observations of the experimenter during the pilot study. Problems which emerged half of the time or more were reported.

(Results Section line 342 - 348)

Feasibility Feedback

In general, participants found the study feasible, however two problems emerged: First, participants reported to have difficulty maintaining a high level of concentration throughout all four conditions and reported getting tired after a while. Second, participants noticed that the videos remained the same in every condition. Some were even able to recall their own decisions on previous videos and this influenced their decisions on the videos in the later conditions. 

Modification based of the pilot study, based on the critiques

Based on the feedback of the participants, we described the modification in the interim discussion (see line: 361ff):

“However, based on the feedback about the difficulty in concentrating and, even more importantly, on recalling their own decision on prior videos, we decided to shorten the video measurement. First, we decided to only measure the SMM for the net player and thus, using only two conditions. Second, as the main study should measure more intuitive decisions rather than deliberate ones, we added a 3-s time limit for ball-taking behavior and movement directions to avoid participants being able to recall their actions and deliberately thinking about their decisions.(cf. [46]). “

---

## [Editor Report · Decision Letter 2]

10 Nov 2020

Do we agree on who is playing the ball? Developing a video-based measurement for Shared Mental Models in tennis doubles

PONE-D-20-06372R2

Dear Dr. Raue,

We’re pleased to inform you that your manuscript has been judged scientifically suitable for publication and will be formally accepted for publication once it meets all outstanding technical requirements.

Kind regards,

Dominic Micklewright, PhD CPsychol PFHEA FBASES FACSM

Academic Editor

PLOS ONE

---

## [Editor Report · Acceptance letter]

19 Nov 2020

PONE-D-20-06372R2 

Do we agree on who is playing the ball? Developing a video-based measurement for Shared Mental Models in tennis doubles 

Dear Dr. Raue:

I'm pleased to inform you that your manuscript has been deemed suitable for publication in PLOS ONE. Congratulations! Your manuscript is now with our production department. 

Kind regards, 

on behalf of

Professor Dominic Micklewright 

Academic Editor

PLOS ONE